# Transcriptomic Analysis of the Differential Nephrotoxicity of Diverse Brominated Flame Retardants in Rat and Human Renal Cells

**DOI:** 10.3390/ijms221810044

**Published:** 2021-09-17

**Authors:** Lillie Marie A. Barnett, Naomi E. Kramer, Amanda N. Buerger, Deirdre H. Love, Joseph H. Bisesi, Brian S. Cummings

**Affiliations:** 1Interdisciplinary Toxicology Program, University of Georgia, Athens, GA 30602, USA; lbarne8@emory.edu (L.M.A.B.); naomi.kramer25@uga.edu (N.E.K.); 2Department of Environmental and Global Health and Center for Environmental and Human Toxicology, University of Florida, Gainesville, FL 32611, USA; abuerger@ufl.edu (A.N.B.); deirdrelove@ufl.edu (D.H.L.); jbisesi@phhp.ufl.edu (J.H.B.J.); 3Department of Pharmaceutical and Biomedical Sciences, University of Georgia, Athens, GA 30602, USA

**Keywords:** brominated flame retardants, Kidney, RNASeq, tetrabromobisphenol A, hexabromocyclododecane, tetrabromodiphenyl ether, nephrotoxicity, gene set enrichment analysis

## Abstract

Brominated flame retardants (BFRs) are environmentally persistent, are detected in humans, and some have been banned due to their potential toxicity. BFRs are developmental neurotoxicants and endocrine disruptors; however, few studies have explored their potential nephrotoxicity. We addressed this gap in the literature by determining the toxicity of three different BFRs (tetrabromobisphenol A (TBBPA), hexabromocyclododecane (HBCD), and tetrabromodiphenyl ether (BDE-47)) in rat (NRK 52E) and human (HK-2 and RPTEC) tubular epithelial cells. All compounds induced time- and concentration-dependent toxicity based on decreases in MTT staining and changes in cell and nuclear morphology. The toxicity of BFRs was chemical- and cell-dependent, and human cells were more susceptible to all three BFRs based on IC_50_s after 48 h exposure. BFRs also had chemical- and cell-dependent effects on apoptosis as measured by increases in annexin V and PI staining. The molecular mechanisms mediating this toxicity were investigated using RNA sequencing. Principal components analysis supported the hypothesis that BFRs induce different transcriptional changes in rat and human cells. Furthermore, BFRs only shared nine differentially expressed genes in rat cells and five in human cells. Gene set enrichment analysis demonstrated chemical- and cell-dependent effects; however, some commonalities were also observed. Namely, gene sets associated with extracellular matrix turnover, the coagulation cascade, and the SNS-related adrenal cortex response were enriched across all cell lines and BFR treatments. Taken together, these data support the hypothesis that BFRs induce differential toxicity in rat and human renal cell lines that is mediated by differential changes in gene expression.

## 1. Introduction

Brominated Flame retardants (BFRs) are organohalogen mixtures that are added to paint, carpets, furniture, electronics, and other household items to reduce their flammability. Because they are not covalently bound to these products, BFRs leach into the environment where they accumulate in dust and animal products that we inhale or ingest [1,2,3,4]. Primary sources of BFR exposure range from dust and breastmilk for infants to fish, meat, and dairy for children and adults [3]. However, several uncertainties remain. For example, although estimated daily intake values are consistently higher in infants relative to other age groups, these values vary depending on geographic location and sampling conditions [2]. Furthermore, although BFRs accumulate in human blood and breastmilk, long-term biomonitoring data are limited for newer BFRs and for populations outside of the European Union. For example, in the US, polybrominated diphenyl ethers (PBDEs) and polybrominated biphenyls (PBBs) are the only BFRs that are monitored by the CDC’s National Health and Nutrition Examination Survey (NHANES). Taken together, these limitations in our understanding of BFR exposure mean that prior BFR risk assessments by the European Union, European Food Safety Authority (EFSA), Health Canada, and US Environmental Protection Agency (US EPA) may be inaccurate.

Despite their marketed purpose of protecting the public, BFRs have become famous for their toxic effects on humans and wildlife [5]. This began in the 1970s when polybrominated biphenyl (PBB)-contaminated animal feed poisoned livestock and caused neurobehavioral deficits in Michigan farming communities [6]. That same decade, tris(2,3-dibromopropyl) phosphate (tris-BP) was banned from childrens’ sleepwear after a mutagenic metabolite was detected in urine [7]. More recently, in 2004, polybrominated diphenyl ethers (PBDEs) were banned in the US after levels in breastmilk were found to double over a span of 2–5 years [8,9]. Consequently, the EPA covers BFRs under its Toxic Substance Control Act, which calls for continued research on the threats that these chemicals pose to public health.

In response to the EPA’s call, a growing body of research links BFRs to toxicity in several target organs, including the kidney. These studies report nephrotoxicity at doses ranging between 1.2–600 mg/kg and at time points ranging from 20 days to 2 years in rats and mice [10,11,12,13,14,15]. However, these studies lack sensitivity, focusing on markers such as kidney weight, histopathology, and serum creatinine levels that do not change until nearly 50% of the nephrons are nonfunctional. Some BFR studies report more sensitive changes in renal oxidative stress in rat kidneys. These include increases in reduced glutathione and thiobarbituric acid reactive substances (TBARS) and decreased total -SH groups after 28-day exposure to BDE-209 at 31.25–500 mg/kg; decreased catalase activity and increased ratio of oxidized: reduced glutathione (GSSG/GSH ratio) after BDE-99 exposure at 1.2 mg/kg; and increased 8-Oxo-2’-deoxyguanosine(8-OHdG) after 30-day exposure to tetrabromobisphenol A (TBBPA) at 250 and 500 mg/kg [16,17,18].

While the above studies demonstrate the nephrotoxic potential of BFRs, the molecular mechanisms that mediate this toxicity are poorly understood. Furthermore, it is unclear whether BFRs induce chemical-dependent and species-dependent toxicity in any target organ, let alone the kidney.

The present study addresses the chemical- and cell -dependence of BFR toxicity in multiple renal cell lines derived from the proximal tubule cells of the kidney. We compared the effects of three BFRs with distinct chemical structures (Table 1). These BFRs include polybrominated diphenyl ether number 47 (BDE-47), which is an older BFR that is banned in the US but remains persistent in the environment; hexabromocyclododecane (HBCD), which is a newer BFR that is still used in the US despite a global ban under the Stockholm Convention on Persistent Organic Pollutants; and tetrabromobisphenol A (TBBPA), which is currently the world’s most highly produced BFR.

This study also explores the molecular mechanism of action of BFR toxicity by measuring transcriptional changes in these cells using RNA sequencing and Gene Set Enrichment Analysis at time points prior to changes in cell morphology and MTT staining. Data from these studies will guide future studies concerning the molecular mechanisms of BFR nephrotoxicity, as well as the use of rodents for assessing the risk of BFRs to human health.

## 2. Results

### 2.1. Dose and Cell Line-Dependent Toxicity of BFRs in Rat and Human Renal Epithelial Cells

We assessed the concentration- and cell-dependent toxicity of BFRs in rat and human renal epithelial cells by measuring MTT staining after 48 h (Figure 1). IC_50_ values in NRK cells were 45 µM, 20 µM, and 60 µM for BDE-47, HBCD, and TBBPA, respectively. IC_50_ values for these respective chemicals were lower in HK-2 cells (7 µM, 5 µM, and 3 µM) and in RPTECs (5 µM, 12 µM, and 13 µM), suggesting that these chemicals have cell line-specific and potentially species-specific toxicity in the kidney. Chemical-dependent differences were also observed in each of these cell lines. Specifically, HBCD exposure resulted in the lowest IC_50_ in NRK cells (20 µM), whereas TBBPA and BDE-47 had the lowest IC_50_ values in HK-2 cells and RPTECs (3 µM and 5 µM), respectively. Decreases in MTT staining were confirmed by studying cell and nuclear morphology at 24 and 48 h of exposure to these IC_50_ values (Figure 2). As was expected, no overt changes in morphology were observed at 24 h. By 48 h, cells in all three cell lines demonstrated similar morphological changes including nuclear condensation, nuclear fragmentation, vacuolization, rounding up, and detachment.

### 2.2. Effect of BFRs on Apoptosis

To determine if BFR-induced changes in MTT staining and cell morphology were due to apoptosis, we measured annexin V and propidium iodide (PI) staining in NRK and RPTEC cells that were treated at the IC_50_ of each BFR (Figure 3). The percentage of early apoptotic (annexin V-positive) NRK cells increased by 13% after TBBPA treatment and by 5% after BDE-47 treatment, whereas HBCD exposure did not cause significant changes. In RPTECs, the percentage of early apoptotic cells increased by 16% after TBBPA treatment and by 11% after HBCD treatment, whereas BDE-47 exposure did not cause significant changes. Interestingly, the chemical with the lowest IC_50_ in each of these cell lines (HBCD and BDE-47 in NRK and RPTECs, respectively) did not cause significant changes in apoptosis. Therefore, we explored the alternative hypothesis that these BFRs decreased MTT staining and altered cell morphology due to a cytostatic effect and found that no BFRs caused significant changes in the cell cycle (Figure 4).

### 2.3. Effect of BFRs on RNA Sequencing and Differential Gene Expression Analysis

To explore the molecular mechanisms that mediate cell- and chemical-specific differences in BFR toxicity, we performed RNA sequencing and differential gene expression analysis in NRK and HK-2 cells after 24 h treatment with each BFR at the 48 h IC_50_ (Appendix A). TBBPA treatment yielded the highest number of differentially expressed genes in HK-2 cells (214) (Figure 5). HBCD treatment yielded the highest number of differentially expressed genes in NRK cells (82). Principal components analysis of the differentially expressed genes suggests that the cell line accounted for 71% of the variability in gene expression. Accordingly, only five differentially expressed genes were shared between cell lines due to TBBPA exposure, and no differentially expressed genes were shared between cell lines exposed to HBCD or BDE-47. Furthermore, only nine genes were differentially expressed across all three BFRs in human cells, and only five genes were differentially expressed across all three BFRs in rat cells, suggesting that these chemicals vary widely in their effects on gene transcription.

### 2.4. Effect of BFRs on Gene Set Enrichment 

Gene set enrichment analysis (GSEA) was performed to predict how BFR effects on gene expression translate to effects on cell function (Appendix A). The coagulation cascade, extracellular matrix turnover, and SNS-related adrenal cortex response were significantly enriched across all cell lines and BFR treatments. The most enriched gene set was eicosanoids in inflammation for all three BFRs in HK-2 cells (Appendix A). The most enriched gene set was estrogen deficiency in female obesity for all three BFRs in NRK cells (Appendix A). The coagulation cascade also appeared in the top 10 enriched pathways for all conditions except TBBPA-treated human cells, although it was still significantly enriched in this treatment group. Furthermore, extracellular matrix turnover appeared in the top 10 enriched pathways for all treatments in NRK cells and was significantly enriched in all treatment groups.

The comparison of enriched gene sets between rat and human cells for each BFR revealed little overlap between cell lines, with the most occurring in HBCD-treated rat and human cells (Figure 6). Specifically, HK-2 and NRK cells shared 6, 20, and 8 enriched gene sets resulting from BDE-47, HBCD, and TBBPA exposure, respectively. Notably, several of these shared gene sets ranked in the top 10 enriched gene sets. For BDE-47, these included extracellular matrix turnover, vitamin A metabolism, and the coagulation cascade. For HBCD, these included CCR1 expression targets, the coagulation cascade, estrogen deficiency in female obesity, extracellular matrix turnover, find me signal: apoptotic cell attracts phagocyte, gastric and pancreatic lipase function, and glycolysis. For TBBPA, these include the coagulation cascade and extracellular matrix turnover.

Enriched gene sets that were shared between TBBPA, HBCD, and BDE-47 were also identified in each cell line (Figure 7). Specifically, 10 and 12 gene sets were enriched by all three BFRs in NRK and HK-2 cells, respectively. In addition to being enriched by all three BFRs, the coagulation cascade, eicosanoids in inflammation, and keep out signal: LTF inhibits neutrophil recruitment and inflammation were ranked in the top 10 enriched gene sets for multiple BFRs in human cells. In fact, eicosanoids in inflammation was the top enriched gene set for all three BFRs. For rat cells, the coagulation cascade, estrogen deficiency in female obesity, extracellular matrix turnover, protein nuclear import and export, and single-strand mismatch were enriched by all three BFRs and ranked in the top 10 enriched gene sets for multiple BFRs. In fact, estrogen deficiency in female obesity was the top enriched gene set for all three BFRs in NRK cells.

### 2.5. Validation of RNASeq by Quantitative Real-Time Polymerase Chain Reaction (qRT-PCR)

RNASeq data were validated by selecting one differentially expressed gene per treatment group and measuring the expression of that gene using qRT-PCR. This was done in both the original samples used for RNA sequencing (Figure 8), as well as in fresh BFR-treated NRK and HK-2 cells (Figure 9A,B). The direction of change for each gene matched between RNASeq and qRT-PCR data, although significant changes were not observed in some cases. Specifically, in NRK cells, mmp13 expression increased after TBBPA and HBCD exposure and srebf1 expression decreased after BDE-47 exposure. In HK-2 cells, alox5ap increased after TBBPA exposure, plat decreased after BDE-47 exposure, and txnip decreased after HBCD exposure. To further validate the RNASeq data and to test their translatability across different cell lines for the same species, alox5ap, plat, and txnip expression were also measured in an additional human tubular epithelial cell line, RPTEC (Figure 9C). The direction of change in expression of these genes also matched the changes observed in HK-2 cells, although they did not reach significance.

## 3. Discussion

This study presents evidence that the molecular mechanisms of BFR nephrotoxicity in renal cells are species- and chemical-dependent. This conclusion is supported by IC_50_s, effects on apoptosis, effects on cell and nuclear morphology, changes in gene expression, and changes in gene set enrichment that differed between BFRs and between rat and human cells. Furthermore, the lower IC_50_s and changes in gene expression that occurred in HK-2 cells repeated in an additional human renal cell line (RPTECs), suggesting that these effects are species-dependent rather than simply cell line-dependent.

This study advances our knowledge of the transcriptional changes that mediate BFR nephrotoxicity. While this study is the first to perform RNA sequencing in BFR-exposed renal cells, prior studies have used RNA sequencing to characterize BFR effects in other target organs. For example, RNA sequencing has been performed in BFR-exposed rats, mice, and zebrafish and in cell lines derived from the brain, liver, lung, and adipose tissue [19,20,21,22,23]. Although these studies did not conduct gene set enrichment analysis (GSEA), there were still several similarities between our GSEA results and the pathways that were altered in these other tissues. Specifically, the enrichment of genes associated with the coagulation cascade, complement cascade, and cytokine-related pathways was reported in BDE-47-treated mouse adipose tissue and in TBBPA-treated human bronchial epithelial cells [19,20]. Additionally, genes involved in the extracellular matrix organization and biological processes related to the cytoskeleton were enriched in TBBPA-treated zebrafish embryos [21]. HBCD exposure also altered glucose and lipid metabolism, as well as fibrotic pathways, in rats [22]. Finally, in mouse neural cells, HBCD altered pathways related to Ca^2+^ and Zn^2+^ signaling, glutamatergic neuron activity, apoptosis, and oxidative stress in vitro and in vivo [23]. Taken together, these similarities highlight common mechanisms that potentially translate across tissues and experimental models.

Despite the similarities described above, differential gene expression analysis of the RNA sequencing data in this study suggested that the transcriptional mechanisms of BFR toxicity differ between chemicals and across human and rat cells. This highlights several important considerations for future mechanistic studies on BFRs and other chemicals. First, because BFR-induced changes in gene expression and gene set enrichment were largely cell-dependent, care should be taken when extrapolating data derived from rats to humans with regards to the ability of BFRs to induce renal dysfunction in humans. This supports the need for studies that compare human and rodent cell lines to identify shared mechanisms that are potentially translatable between the two. Second, because BFR-induced changes in gene expression and gene set enrichment were chemical-dependent, this supports the need for future studies that explore how differences in BFR chemical structure produce specific mechanisms of toxicity.

Despite the cell- and chemical-dependent effects on toxicity and gene expression noted above, BFRs shared some enriched gene sets. For example, TBBPA, HBCD, and BDE-47 shared the same top enriched gene set in each cell line. Specifically, estrogen deficiency in female obesity was the top enriched gene set for all three BFRs in rat kidney cells, and eicosanoids in inflammation was the top enriched gene set for all three BFRs in HK-2 cells. Furthermore, eicosanoids in inflammation was enriched exclusively in HK-2 cells, suggesting that this is an important mechanism behind the species-specific effects of BFRs in human cells. Future research should explore whether these shared effects are tied to any subtle similarities in the chemical structures of these BFRs. Such studies could aid the development of safer alternative flame retardants.

These findings support prior research on the effects of BFRs in other target organs. Both TBBPA and BDE-47 have been shown to increase prostaglandin synthesis in human trophoblasts [24,25]. Because eicosanoids including prostaglandins regulate trophoblast processes that are essential for placentation and pregnancy, this is an important mechanism to consider for studies on the developmental and reproductive toxicity of BFRs.

TBBPA, HBCD, and BDE-47 have been shown to enhance diet-induced weight gain in vivo and to alter adipogenesis, adipocyte differentiation, and lipid metabolism in mouse preadipocytes [26,27,28,29,30,31,32,33]. However, no study has explored the potential involvement of estrogen signaling in the obesogenic effects of these BFRs. Interestingly, all three of these BFRs have been shown to have anti-estrogenic activity, either directly through ER antagonism (as is the case for HBCD and OH-BDE-47) or indirectly through receptor-independent mechanisms (as is the case for TBBPA) [34,35,36]. Furthermore, prior studies implicate a role for certain genes within the “estrogen deficiency and female obesity” GSEA gene set, including PPARγ, in the mechanism of adipocyte differentiation for all three of these chemicals [30,32,33,37]. Because the estrogen receptor impacts adipogenesis via effects on PPARγ, the endocrine disrupting effects of BFRs on these receptors are important mechanisms to consider for studies on the obesogenic effects of these BFRs [38].

This study is the first to identify GSEA functional gene sets that were enriched across both rat and human cell lines for BFRs. Notably, several of these shared gene sets ranked in the top 10 enriched gene sets for a given BFR, making them both mechanistically relevant and potentially translatable across human and rodent models.

Interestingly, some of these species-independent GSEA functional sets were shared across all three BFRs studied. These include extracellular matrix turnover, the coagulation cascade, and the SNS-related adrenal cortex response. No studies have explored the effects of TBBPA, HBCD, or BDE-47 on blood coagulation or on coagulation factors. Future studies should explore these effects in vivo.

TBBPA and BDE-47 have been previously shown to alter matrix metalloproteinases (MMPs), which function in the “extracellular matrix turnover” GSEA pathway. Specifically, TBBPA-induced alterations in MMP expression accompanied developmental effects in zebrafish embryos and breast cancer cell metastasis [39,40]. BDE-47-induced alterations in MMP expression were linked to impaired human trophoblast migration and invasion (which is a potential mechanism of pregnancy disorders) and to neuroblastoma metastasis [41,42]. Because MMPs mediate tissue morphogenesis, wound healing, and cell migration, more studies are warranted that explore how disrupted extracellular matrix turnover contributes mechanistically to BFR effects on development, pregnancy, and cancer.

Regarding the enrichment of genes related to the SNS-related adrenal cortex response, a few studies have explored the potential of BFRs to interfere with hormones released by the adrenal gland, such as cortisol. Specifically, TBBPA has antagonistic effects on glucocorticoid receptor signaling, and this has been linked to its obesogenic effects [32,33,43]. Our data suggest that a similar response may occur in the kidney. They also support the need for more research on this mechanism and whether it also plays a role in HBCD and BDE-47 induced obesity and metabolic dysfunction.

Several of the enriched gene sets in this study have been suggested to mediate renal dysfunction. Eicosanoids and arachidonic metabolism play major roles in kidney inflammation and have been shown to predict chronic kidney disease progression in patients [44,45,46]. Extracellular matrix turnover plays a role in aberrant wound healing and fibrosis, which is a hallmark of kidney disease. Accordingly, matrix metalloproteinases and their tissue inhibitors (TIMPs) have also been associated with chronic kidney disease progression [47,48,49,50,51,52]. Finally, hypercoagulability is a major complication during the advanced stages of kidney disease, and coagulation proteases have been shown to predict chronic kidney disease progression in patients [53,54,55,56]. Several of these putative biomarkers of kidney injury were differentially expressed by BFRs according to our RNASeq data and have never previously been associated with BFR toxicity. Specifically, arachidonate 5-lipoxygenase activating protein (alox5ap) and tissue plasminogen activator (plat) have not been studied in relation to any BFR. Matrix metalloproteinases (MMPs) were altered by both TBBPA and BDE-47 exposure in trophoblasts; however, they have never been related to BFR toxicity to the kidney.

It is important to note that this study is limited because the acute exposure time and relatively high doses used may not reflect typical human exposure to BFRs. There is a great deal of uncertainty regarding the levels of BFRs in humans. Few studies report levels outside of the United States, European Union, and China. Furthermore, within the US, BDE-47 is the only BFR that has been measured via the NHANES survey. While this limitation was addressed by conducting RNASeq analysis at 24 h (in the presence of minimal toxicity), further studies are needed that validate these findings after chronic exposure. Additionally, for the purpose of interpreting the GSEA results, the “top 10” gene sets could be ranked according to the strength of their *p* values rather than according to effect size (median log FC in expression). However, this did not impact which GSEA results appear in the top 10 pathways. Furthermore, given that the GSEA results already have a threshold *p*-value of <0.05, we chose to rank them according to effect size. Finally, these data need to be validated in vivo. Our future studies will address these limitations by exploring the potentially shared mechanisms identified by GSEA in vivo after long-term exposure to lower doses that approximate human exposure. Such studies will have limitations given that our data suggest that BFR-induced transcriptional changes are largely species-dependent. Regardless, the data reported in this study set a solid foundation for future in vivo studies investigating the molecular mechanisms of BFR nephrotoxicity. These studies will substantially increase our knowledge about the risks that these persistent organic pollutants pose to human health.

## 4. Materials and Methods

### 4.1. Chemicals and Reagents

3,3′,5,5′-Tetrabromobisphenol-A (TBBPA), 1,2,5,6,9,10-Hexabromocyclododecane (HBCD), and 2,2′,4,4′-Tetrabromodiphenyl ether (BDE-47) were purchased as powder from Sigma (St. Louis, MO, USA). Stock solutions of each BFR were dissolved in DMSO (Fisher Scientific, Fair Lawn, NJ, USA). MTT [3-(4, 5-dimethylthiazol-2-yl)-2, 5-diphenyltetrazolium bromide] was purchased from Sigma (St. Louis, MO, USA) and dissolved in PBS (Fisher Scientific, Fair Lawn, NJ, USA). DMEM, DMEM F-12, Keratinocyte Serum-Free media, human recombinant epidermal growth factor, bovine pituitary extract, hTERT RPTEC Growth Kit, and penicillin/streptomycin were purchased from American Type Culture Collection (ATCC, Manassas, VA, USA). Fetal bovine serum was purchased from VWR (Radnor, PA, USA). G418 solution was purchased from Fisher Scientific (Fair Lawn, NJ, USA). 4% Paraformaldehyde solution was purchased from Sigma (St. Louis, MO, USA). Hoechst 33342 was purchased from Sigma (St. Louis, MO, USA) and dissolved in water. Primers were purchased from Integrated DNA Technologies (Newark, NJ, USA). All-in-One ™ qPCR Mix was purchased from GeneCopoeia (Rockville, MD, USA). SsoAdvanced™ Universal SYBR Green Supermix was purchased from Bio-Rad (Hercules, CA, USA). Superscript™ IV VILO™ Master Mix was purchased from Thermo Fisher Scientific (Waltham, MA, USA). The NEBNext Ultra II RNA Library Prep Kit with Sample Purification Beads, NEBNext Library Quantification Kit, NEBNext Multiplex Oligos for Illumina, and NEBNext Poly(A) mRNA Magnetic Isolation Module were purchased from New England Biolabs (Ipswich, MA, USA).

### 4.2. Cell Culture and Treatment with BFRs

Rat (NRK 52-E) and human (HK-2, RPTEC) kidney cells were purchased from ATCC (Manassas, VA, USA) and maintained according to the vendor’s recommendations. NRK 52-E cells were maintained in DMEM supplemented with 10% fetal bovine serum and 1% penicillin/streptomycin. HK-2 cells were maintained in Keratinocyte Serum-Free media supplemented with 2.5 µg human recombinant epidermal growth factor, 25 mg bovine pituitary extract, and 1% penicillin/streptomycin. RPTECs were maintained in DMEM F-12 and supplemented with hTERT RPTEC Growth Kit and 0.1 mg/mL G418 solution. All cell lines were kept in an incubator at 37 °C in 5% CO_2_ and were given media changes every 2–3 days. When cells were plated for experiments, they reached 80% confluence prior to treatment with TBBPA, HBCD, BDE-47, or an equivalent volume of DMSO vehicle. The volume of DMSO did not exceed 0.5% of the total volume in each well.

### 4.3. MTT Assay and Calculation of IC_50_ Values

NRK, HK-2, and RPTEC cells were seeded in 48-well tissue culture plates at 5 × 10^4^ cells per well and were treated for 48 h with 1 µM, 3 µM, 10 µM, 30 µM, and 100 µM of TBBPA, HBCD, BDE-47, or an equivalent volume of DMSO. After BFR exposure, cells were incubated for 2 h with MTT dye at a final concentration of 0.25 mg/mL. Nonreduced MTT and media were then aspirated and replaced with DMSO to dissolve the MTT formazan crystals. Plates were shaken for an additional 15 min and absorbance was read at 570 nm using a Spectra Max M2 plate reader (BMG Lab Technologies, Cary, NC, USA). Absorbance values were averaged across triplicate wells for each dose of each BFR and were normalized to the average absorbance from DMSO-treated wells. The IC_50_ for each BFR in each cell line was calculated in Graphpad Prism 8 (San Diego, CA, USA) using the [Inhibitor] vs. response–Variable slope model.

### 4.4. Cell and Nuclear Morphology

NRK, HK-2, and RPTEC cells were seeded in 12-well tissue culture plates at 150,000 cells per well. Cells were treated for 24 h and 48 h with each BFR at the IC_50_ calculated from the MTT assay or with an equivalent volume of DMSO. Cell morphology was assessed using a Nikon Eclipse Ti-U inverted microscope (Melville, NY, USA). Cells were then fixed in 4% paraformaldehyde and stained with 1 µg/mL Hoechst 3,3342 for 15 min in the dark at room temperature. Pictures were taken using the same microscope with a DAPI fluorescence filter. 

### 4.5. Measurement of Cell Death via Annexin V and PI Staining 

Cells were seeded in 12-well tissue culture plates at 150,000 cells per well. Cells were treated for 24 h with each BFR at the IC_50_ calculated from the MTT assay or with an equivalent volume of DMSO. Cells were harvested via trypsinization and stained with Annexin V-Alexa Fluor^®^ 488 and PI using the Tali Apoptosis kit (Life Technologies, Carlsbad, CA, USA) according to the manufacturer’s protocol. Annexin V and PI fluorescence were measured using a NovoCyte Quanteon flow cytometer (Agilent, Santa Clara, CA, USA) at the University of Georgia’s Cytometry Shared Resource Laboratory. Cells were run at a flow rate of <500 events/second. The resulting scatterplots were analyzed using FlowJo version 10.6.2 (BD Biosciences, San Jose, CA, USA). Events above 100,000 scatter units on an FSC vs. SSC plot were considered debris and eliminated from the analysis. The percentage of cells in early apoptosis (Annexin V-positive), late apoptosis (Annexin V and PI-positive), and necrosis (PI-positive) was averaged across duplicate wells for each BFR and across n = 3 passages of each cell line.

### 4.6. Cell Cycle Analysis via Propidium Iodide Staining

Cells were seeded in 12-well tissue culture plates at 150,000 cells per well. Cells were treated for 24 h with each BFR at the IC_50_ calculated from the MTT assay or with an equivalent volume of DMSO. Cells were harvested via trypsinization and fixed for 2 h at 4 °C in 70% ethanol. Cells were washed twice with PBS and incubated at room temperature for 5 min with 100 μg/mL DNAse-free ribonuclease (Sigma, St. Louis, MO, USA). This solution was diluted 1:5 in a 50 μg/mL PI staining solution (Sigma, St. Louis, MO, USA) and incubated in the dark overnight at 4 degrees. PI fluorescence was measured at 605 nm using a NovoCyte Quanteon flow cytometer (Agilent, Santa Clara, CA, USA) at UGA’s Cytometry Shared Resource Laboratory. Cells were run at a flow rate of <500 events/second. Cell cycle analysis was performed using FlowJo version 10.6.2 (BD Biosciences, San Jose, CA, USA). Events above 100,000 scatter units on an FSC vs. SSC plot were considered debris and eliminated from the analysis. Doublets were also discriminated from analysis using a height vs. area plot. The percentage of cells in the G0/M, S, and G2 phases was averaged across duplicate wells for each BFR and across n = 3 passages of each cell line.

### 4.7. RNA Isolation, Sequencing, and Bioinformatic Analysis

For RNA sequencing experiments, cells were treated for 24 h with each BFR at the 48 h IC_50_ calculated from the MTT assay or with an equivalent volume of DMSO. Total RNA was isolated from five different passages for each cell line. Cells were homogenized in TRIzol^TM^ Reagent (Fisher Scientific, Waltham, MA, USA). Total RNA was extracted with chloroform and purified with the RNeasy Mini Kit (Qiagen, Hilden, Germany) using the manufacturer’s protocol. The quantity and quality of RNA was examined using a Thermo Scientific^TM^ NanoDrop^TM^ 8000 Spectrophotometer (Waltham, MA, USA). The integrity of RNA was examined using an Agilent 2100 Bioanalyzer (Agilent Technologies, Santa Clara, CA, USA). Only RNA with OD 260/280 ≥ 1.8 and RNA integrity number ≥ 7 was used for subsequent experiments. Samples were quantified again using the Qubit 3.0 fluorometer (ThermoFisher Scientific, Waltham, MA, USA). Libraries were prepared using the NEBNext Ultra II RNA Library Prep Kit for Illumina (New England BioLabs, Ipswich, MA, USA) in combination with the NEBNext Poly(A) magnetic isolation module according to the manufacturer’s directions. Sample libraries were pooled and quantified before sequencing using the NEBNext Library Quant Kit for Illumina (New England Biolabs, Ipswich, MA, USA). Sequencing was conducted at the University of Florida Interdisciplinary Center for Biotechnology Research using the Illumina Nova Seq 6000 system (Illumina, San Diego, CA, USA). Raw reads were demultiplexed, QCed- and trimmed using FastQC. Low-quality reads were removed using Trimmomatic [57]. Reads from HK-2 cells were mapped to the *Homo sapiens* genome (GRCm38) and reads from NRK cells were mapped to *Rattus norvegicus* genome (Rnor_6.0) using Star Aligner [58]. Reads were normalized as counts per millon (CPM), gene expression was calculated using RSEM and analyzed for differential gene expression using EdgeR [59,60]. Significant up-and down-regulated genes were selected based on the following criteria: (1) the fold change was greater than 2 and (2) the false discovery rate (FDR) adjusted q-value was less than 0.05. To further understand the mechanistic functions of the differentially expressed genes, gene set enrichment analysis (GSEA) was conducted in PathwayStudio^TM^ (Elsevier, Amsterdam, the Netherlands) using gene sets involved in Biological Processes, Cell Processes, and Toxicity Pathways. Gene sets with *p* < 0.05 were significantly enriched. The top 10 significantly enriched gene sets were ranked according to the absolute value of the median log FC in expression for each gene set (reported as “mean change” in the GSEA output.).

### 4.8. Validation via qRT-PCR

Five representative genes (one per treatment for each cell line) were selected from the RNASeq data for validation via quantitative reverse-transcriptase PCR (qRT-PCR). All gene specific primers used in this study were designed using Primer 3 software and were validated for specificity via gel electrophoresis (Table 2). qRT-PCR was conducted in the original samples from RNA sequencing, as well as in RNA from freshly treated cells. Freshly treated cells were exposed to BFRs for 24 h at the same concentrations that were used for RNASeq. RNA was extracted from freshly treated cells as previously described above and converted to cDNA using the Superscript™ IV VILO™ Master Mix according to the manufacturer’s protocol. qRT PCR was performed on a QuantStudio 5 system (Applied Biosystems, Foster City, CA, USA) with All-in-One ™ qPCR mix for fresh RNA samples. The CFX Connect system (Bio-Rad, Hercules, CA, USA) and SsoAdvanced™ Universal SYBR Green Supermix were used for qRT PCR original RNASeq samples. All qRT-PCR experiments were carried out using the kit-recommended cycling parameters and a T_a_ of 57 °C for all primers. Also, 50 ng cDNA and 0.2 µM primers were included in each qRT-PCR reaction (10 µL total). Gene expression was calculated as a fold change relative to DMSO-treated cells using the 2^−ΔΔCt^ method. GAPDH and β-actin were used as internal controls for normalization.

### 4.9. Statistical Analysis

Statistical analyses were performed using the GraphPad Prism software Version 9 (San Diego, CA, USA). Data were analyzed for normality and subsequently, either a parametric or non-parametric test was used to determine statistical significance. In cases where multiple BFRs were compared simultaneously to vehicle control, one-way ANOVA was used with Dunnett’s post hoc test. In cases where only one BFR was compared to vehicle control, an unpaired *t*-test was used.

## 5. Conclusions

This study compares the molecular mechanisms of BFR toxicity in rat and human kidney cells after acute exposure to three BFRs for the first time. RNA sequencing and differential gene expression analysis suggested that BFRs have chemical- and species-dependent effects on gene expression. However, the GSEA results highlighted several functional changes that are potentially shared between rats and humans for all three BFRs, including the coagulation cascade, extracellular matrix turnover, and the SNS-related adrenal response. Several of the top enriched gene sets are involved in kidney disease progression and provide insight into the mechanisms of BFR-induced kidney injury. Others, while not relevant to kidney disease, still identify potential mechanisms of BFR-induced developmental toxicity, reproductive toxicity, and obesity. Future studies will explore these potential mechanisms in vivo at environmentally relevant exposures to BFRs.

## Figures and Tables

**Figure 1 ijms-22-10044-f001:**
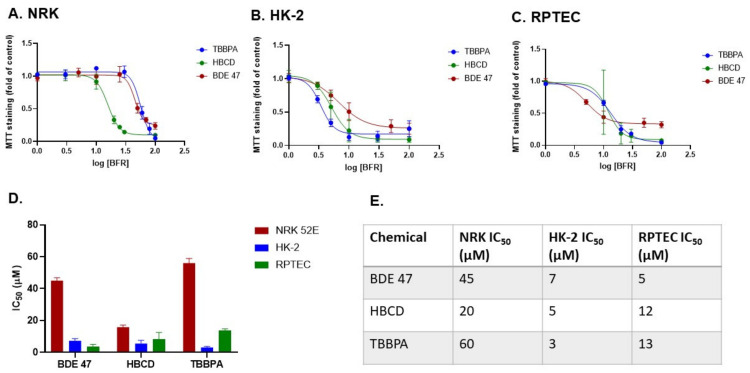
Effect of BFRs on MTT staining and resulting IC_50_ values in rat vs. human cells. (**A**) NRK, (**B**) HK-2, and (**C**) RPTEC cells were exposed for 48 h to TBBPA, HBCD, and BDE-47 at a range of concentrations (1–100 µM) or an equivalent volume of DMSO vehicle. (**D**,**E**) Mean IC_50_ values were calculated across dose-response curves for *n* = 5 passages of cells.

**Figure 2 ijms-22-10044-f002:**
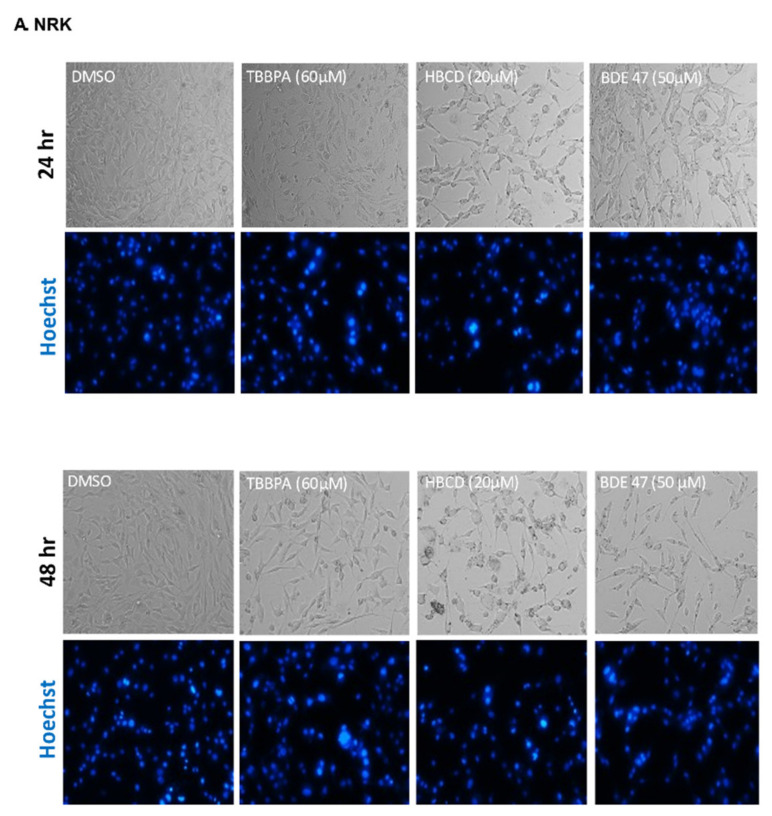
Effect of BFRs on cell and nuclear morphology in (**A**) NRK (**B**) HK-2 and (**C**) RPTECs after 24 and 48 h of exposure to the IC_50_. Data are representative of at least 3 (*n* = 3) separate experiments on separate passages.

**Figure 3 ijms-22-10044-f003:**
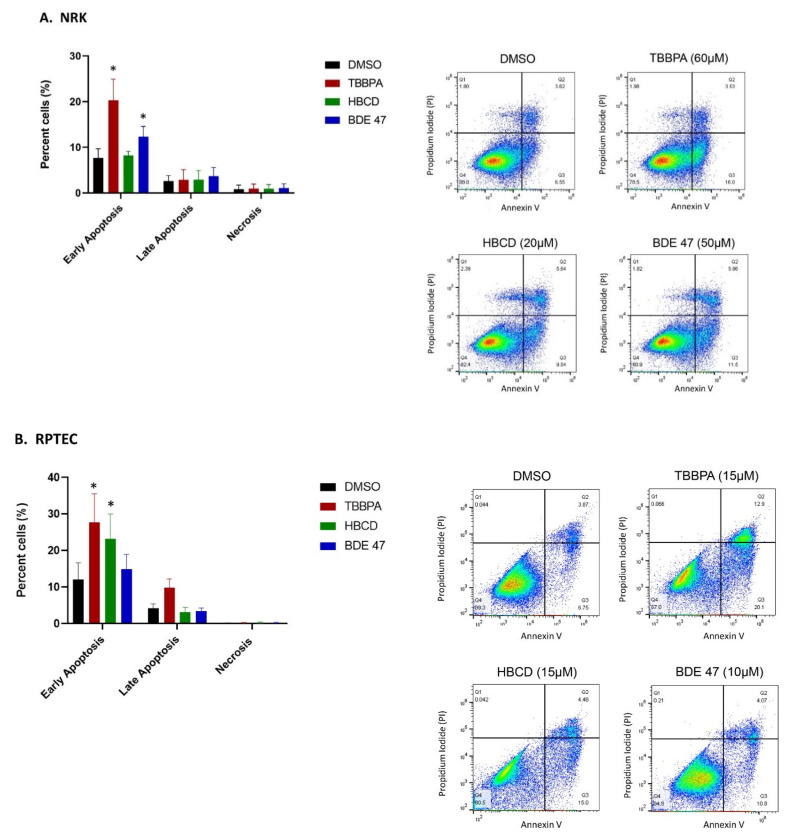
Effect of BFRs on annexin V and propidium iodide staining in (**A**) NRK cells and (**B**) RPTECs after 24 hours of exposure to the IC_50._ Each bar represents the mean ± SD percentage of cells undergoing each phase of apoptosis or necrosis across *n* = 3 passages of cells. Scatterplots show the percentage of cells in each phase of apoptosis for one representative passage and were used to generate the corresponding bar graphs. * Indicates significant difference (*p* < 0.05) compared to DMSO control.

**Figure 4 ijms-22-10044-f004:**
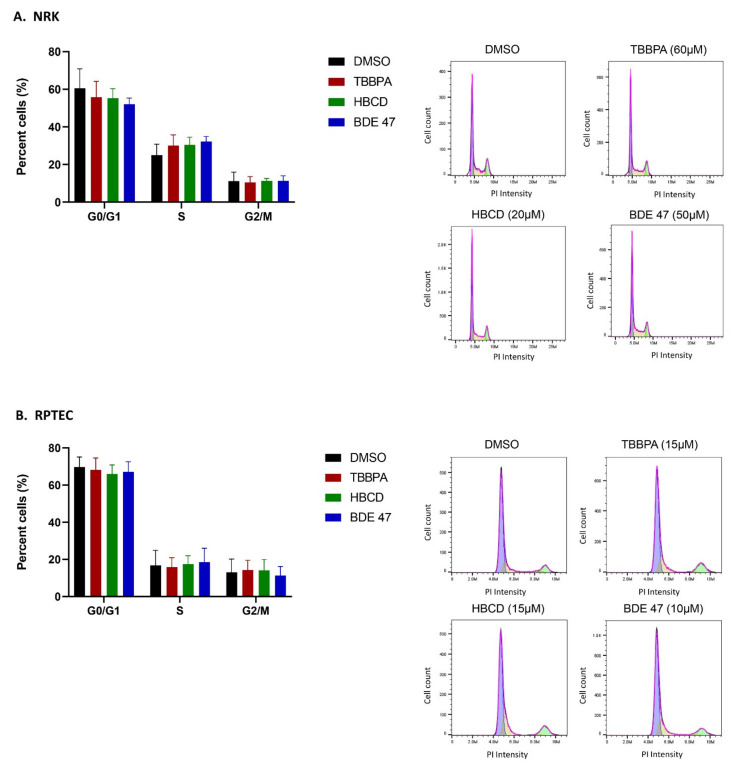
Effect of BFRs on the cell cycle in (**A**) NRK cells and (**B**) RPTECs after 24 h of exposure to the IC_50._ Each bar represents the mean ± SD percentage of cells undergoing each phase of the cell cycle across *n* = 3 passages of cells. Histograms show the number of cells in each phase of the cell cycle for one representative passage and were used to generate the corresponding bar graphs.

**Figure 5 ijms-22-10044-f005:**
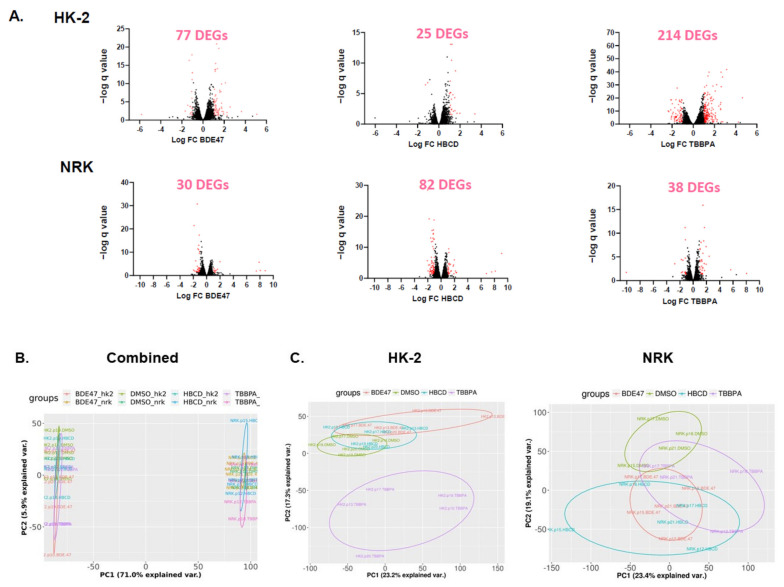
Differential gene expression analysis of RNASeq data in BFR-treated rat (NRK) and human (HK-2) cells. Cells were exposed for 24 h to each BFR at the 48-h IC_50s_ shown in Figure 1. This was repeated across *n* = 5 passages for each cell line. (**A**) Volcano plots depict the number of differentially expressed genes for each treatment group. Significantly up-and down-regulated genes are shown in pink and were selected based on the following criteria: (1) the fold change was >2 and (2) the false discovery rate (FDR) was < 0.05. (**B**,**C**) Principal components analysis (PCA) of the significantly differently expressed genes (DEGs) from (**B**) all treatment groups combined and (**C**) within each cell line. Each color represents a specific cell line that was treated with a specific BFR according to the color legend above each PCA plot. (**D**) Venn diagrams showing the overlap in DEGs between rat and human cells for each BFR. (**E**) Venn diagrams showing the overlap in DEGs between BFRs for rat and human cells, respectively.

**Figure 6 ijms-22-10044-f006:**
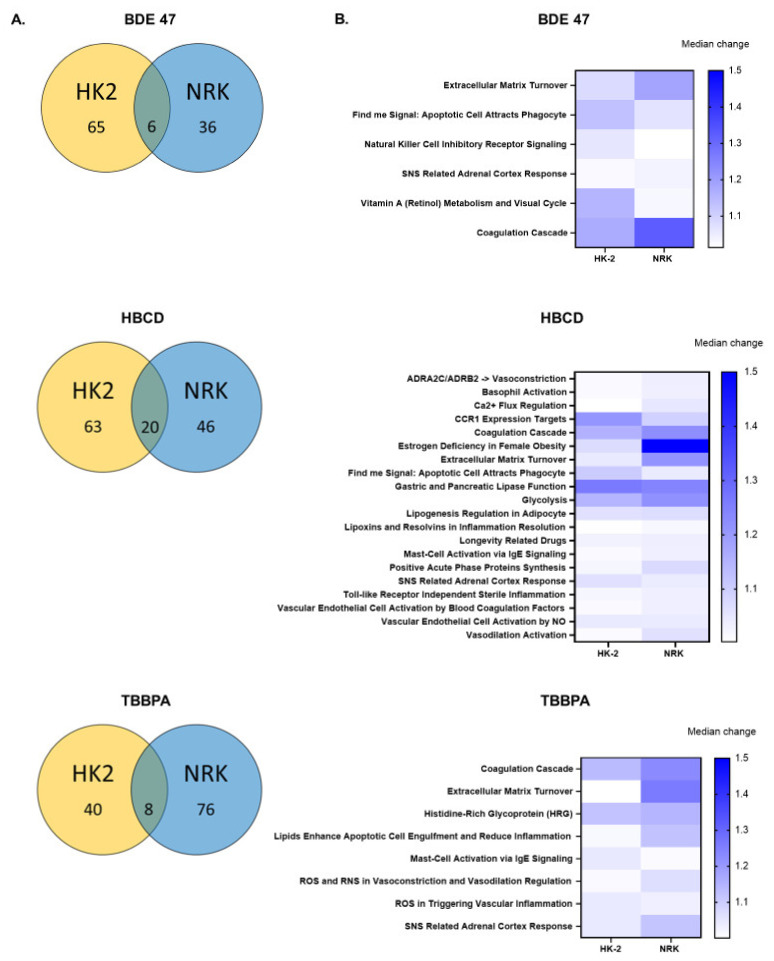
Gene set enrichment analysis of RNASeq data: Comparison between rat (NRK) and human (HK-2) cells. (**A**) Venn diagrams showing the overlap in enriched gene sets between rat vs. human cells. (**B**) Heat maps depict the absolute value of the median log FC for enriched gene sets that were shared between both cell lines for each chemical.

**Figure 7 ijms-22-10044-f007:**
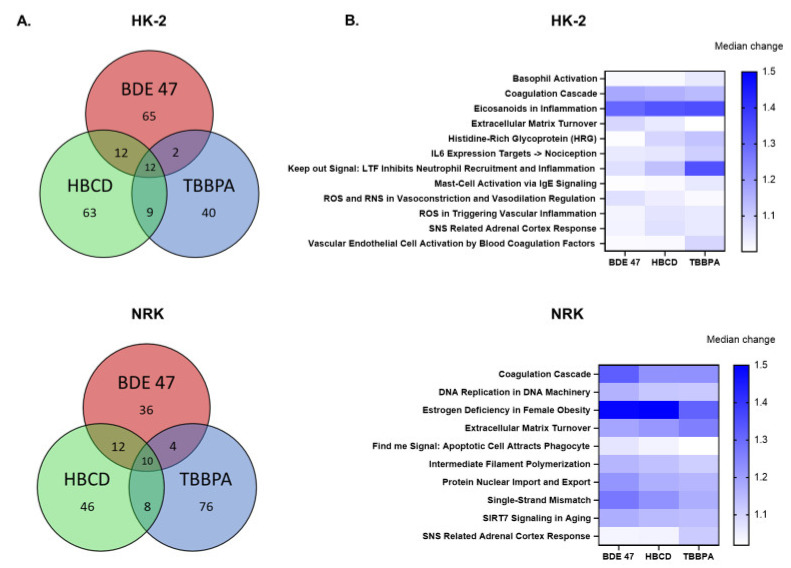
Gene set enrichment analysis of RNASeq data: Comparison between BFRs. (**A**) Venn diagrams showing the overlap in enriched gene sets between BFRs in each cell line. (**B**) Heat maps depict the absolute value of the median log FC for enriched gene sets that were shared between all 3 BFRs in rat and human cell lines, respectively.

**Figure 8 ijms-22-10044-f008:**
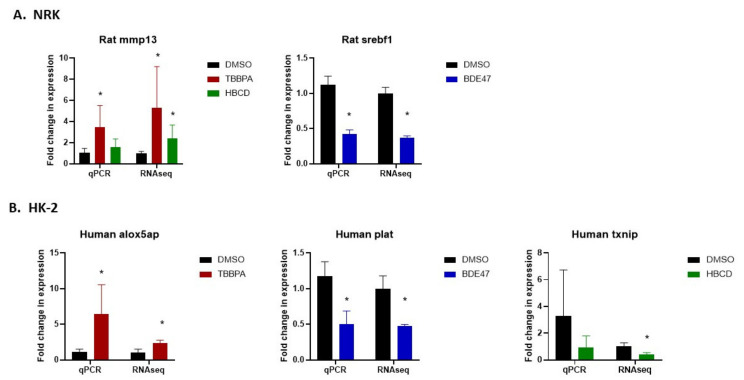
Validation of RNAseq data using qRT-PCR for select differentially expressed genes in the original samples from RNA sequencing. (**A**) Comparison between delta-delta ct values from qRT-PCR and fold-change in CPM values from RNASeq in NRK cells for mmp 13 in TBBPA and HBCD-treated cells and for srebf1 in BDE-47-treated cells. (**B**) Comparison between delta-delta ct values from qRT-PCR and fold-change in CPM values from RNAseq in HK-2 cells for alox5ap in TBBPA-treated cells, for plat in BDE-47-treated cells, and for txnip in HBCD-treated cells. GAPDH was used as a housekeeping gene for all qRT-PCR experiments. Each bar represents the mean ± SD across *n* = 3–5 passages of cells. * Indicates significant difference (*p* < 0.05) compared to DMSO control.

**Figure 9 ijms-22-10044-f009:**
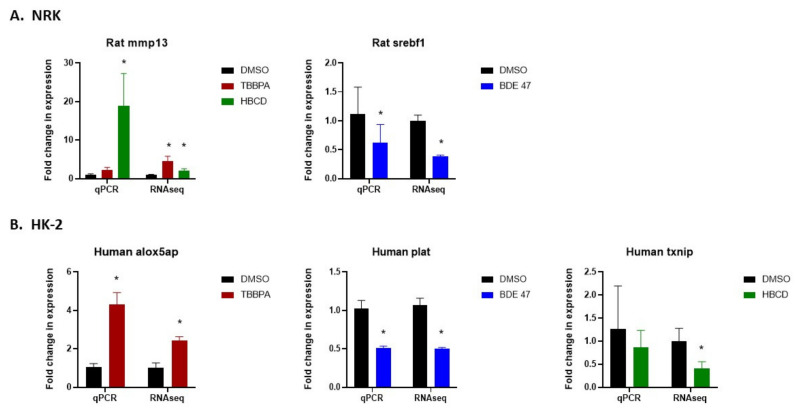
Validation of RNAseq data using qRT-PCR for select differentially expressed genes in freshly treated cells after 24 h exposure to BFRs. (**A**) Comparison between delta-delta ct values from qRT-PCR and fold-change in CPM values from RNAseq in NRK cells for mmp 13 in TBBPA and HBCD-treated cells and for srebf1 in BDE-47-treated cells. (**B**) Comparison between delta-delta ct values from qRT-PCR and fold-change in CPM values from RNAseq in HK-2 cells for alox5ap in TBBPA-treated cells, for plat in BDE-47-treated cells, and for txnip in HBCD-treated cells. (**C**) qRT-PCR data in RPTECs for the same genes as in (**B**). GAPDH was used as a housekeeping gene for all qRT-PCR experiments. Each bar represents the mean ± SD across *n* = 3–5 passages of cells. * Indicates significant difference (*p* < 0.05) compared to DMSO control.

**Table 1 ijms-22-10044-t001:** Name and Structure of Brominated Flame Retardants Used in this Study.

Name	Chemical Structure
2,2’,4,4’-Tetrabromodiphenyl ether	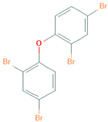
1,2,5,6,9,10-Hexabromocyclododecane	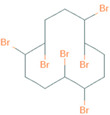
3,3’,5,5’-Tetrabromobisphenol A	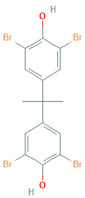

**Table 2 ijms-22-10044-t002:** Primer Sequences Used in this Study.

Gene Name	Species	Forward Sequence	Reverse Sequence
*Alox5ap*	Mouse	TCACCCTCATCAGCGTGGTC	CATCTACGCAGTTCTGGTTGGC
	Human	CCGGAACACTTGCCTTTGAGC	CAAACGCAGCAGGAACTTGGC
*β-actin*	Rat	ACAACCTTCTTGCAGCTCCTCC	TGACCCATACCCACCATCACAC
	Mouse	TATAAAACCCGGCGGCGCAA	TCGTCATCCATGGCGAACTGG
	Human	AGCTCACCATGGATGATGATATCGC	ATAGGAATCCTTCTGACCCATGCC
*Gapdh*	Rat	CAGTGCCAGCCTCGTCTCATA	ACTGTGCCGTTGAACTTGCC
	Human	CTCCTGTTCGACAGTCAGCC	GCCCAATACGACCAAATCCGT
*Mmp 13*	Rat	GCATACGAGCATCCATCCCG	AAGAGGGTCTTCCCCGTGTC
	Mouse	TCCCTAGGTCTGGATCACTCC	TTAGGGTTGGGGTCTTCATCG
*Plat*	Mouse	TGCAGGAACTCAAGAGCTCAG	TTTCTTCTGTGTCCCGAGAGTG
	Human	GTCTTCGTTTCGCCCAGCCA	CGGTTGCTTCTGAGCACAGGG
*Srebf1*	Rat	GCTGCAGGAAACTGAGAGACC	TGGGCTGGATTCCACCTTTC
*Txnip*	Human	GCAGTGCAAACAGACTTCGG	CATCTCATTCTCACCTGTTGGC

## Data Availability

Raw sequencing data from the RNAseq experiments will be submitted to the Gene Expression Omnibus (GEO) and the accession number will be available upon publication.

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
