# Peer review of "Transcriptomic Analysis of the Differential Nephrotoxicity of Diverse Brominated Flame Retardants in Rat and Human Renal Cells"

_ijms, 2021, doi:10.3390/ijms221810044_

Round 1
Reviewer 1 Report
The manuscript is well-conducted and the research results are very interesting, considering the great impact of this compounds over the environment. However, there are a minor revisions required to improve this manuscript.
Introduction section:
Lines 65-72: the three compounds analyzed are not well described. Please, add more information about the main uses of each compound, levels in the environment (water, soil, animals..) and, levels of exposure to human (maybe, through consumption of a food group... There is any limitation of any of this compounds ?? Please, add legislation about limitations of this compounds.
Author Response
Point 1: Lines 65-72: the three compounds analyzed are not well described. Please, add more information about the main uses of each compound, levels in the environment (water, soil, animals..) and, levels of exposure to human (maybe, through consumption of a food group... There is any limitation of any of this compounds ?? Please, add legislation about limitations of this compounds.
Response 1: Thank you for this valuable comment. We have added more information on these compounds to the introduction. This includes a brief description of levels in the environment and estimated levels and sources of exposure to humans. The legislation limits are mentioned. As mentioned, this is brief. Summarizing all these studies would unduly increase the length of the introduction and give it the look of review. Furthermore, data on the levels of these compounds varies widely across different studies and geographic locations, so again, providing precise numbers that are representative of the general population is a challenge that is beyond the scope of this manuscript. However, we have addressed the reviewer’s concerns by citing several recent reviews with this information.
Reviewer 2 Report
Reviewer’s comments
Title: Transcriptomic Analysis of the Differential Nephrotoxicity of Diverse Brominated Flame Retardants in Rat and Human Renal Cells
Journal: International Journal of Molecular Sciences
The manuscript entitled “Transcriptomic Analysis of the Differential Nephrotoxicity of Diverse Brominated Flame Retardants in Rat and Human Renal Cells” describes the potential using of three different BFRs in rat (NRK 52E) and human (HK-2 and RPTEC) tubular epithelial cells. The work design an complimentary approach in achievement of the main goal of study. Moreover, adequate investigations have been performed regarding the mechanism mediations. However, to the present research need strong improvement in order to allow the reader to follow easier the manuscript without difficulties and misunderstanding. This aspect will lead to show the impact of three compounds to advised our knowledge in the transcription changes and not only. Therefore, please follow step by step the comments bellow:
- The results should be highly improved. The figures should be clearly described. The obtained results are poorly described; when the results are presented the authors should include also the obtained value not only write about the effect: lower increasing or higher. Also the results section should include the presentation of the results concerning the values (mean IC50 etc.) then, the noticed observation (means this value increase or decrease). What I would like to underline is the fact that the interpretation of the results should start firstly, with the what the respective figure illustrate and then, about the presentation of the results (means obtained values) and after, the comparison/description (interpretation). Please provide to the results section the required corrections in order to allow the reader to follow the idea. The results are presented more as form of discussion.
- Based on the valuable results, is mandatory to design both: the scheme representing shortly step by step the present work as well as the mechanism mediation on NRK 52E and human HK-2 and RPTEC (in vitro). This aspect will support the main hypothesis proposed in present study. Then, when the author when will prepare the additional studies will design the mechanism in vivo.
- material and methods section: please supplement the Flow cytometry study with the used parameters (means cells/sec., etc. ). Also the information regarding the threshold. Did the author performed the threshold step? How the cells of your interest have been measured using fluorophores for investigation of early, late apoptosis and necrosis (means the way from the receiving ssc/fsc to characteristic scatterplot). Have been excluded from the results interpretation the scattered plots that are not of your interest? Please provide and explanation to the materials section
- table 1: the quality of the compound structures from the table should be improved.
- Figures 1, 3-5, 8, 9: the resolution of the figures must be improved. Especially, it is difficult to recognize the axis x/y name from the histograms received from the flow cytometry analysis. Also Figure 5, the statistical data are really unreadable. Please not remove but improve. All represented info is valuable.
- Legends also should include clearly the information what represent. For example Fig 4: “Effect of BFRs on the cell cycle in (a) NRK cells and (b) RPTECs after 24 hours of exposure to the IC50. Each bar represents the mean ± SD percentage of cells undergoing each phase of the cell cycle across n=3 passages of cells” – the respective legend did not present clearly what illustrate. Should be include representation of the bar graph as well as the histograms, representing what?......in this context please revise all figures legends.

Author Response
Point 1: The manuscript entitled “Transcriptomic Analysis of the Differential Nephrotoxicity of Diverse Brominated Flame Retardants in Rat and Human Renal Cells” describes the potential using of three different BFRs in rat (NRK 52E) and human (HK-2 and RPTEC) tubular epithelial cells. The work design an complimentary approach in achievement of the main goal of study. Moreover, adequate investigations have been performed regarding the mechanism mediations.
Response 1: Thank you for the kind comments.
Point 2: The results should be highly improved. The figures should be clearly described. The obtained results are poorly described; when the results are presented the authors should include also the obtained value not only write about the effect: lower increasing or higher. Also the results section should include the presentation of the results concerning the values (mean IC50 etc.) then, the noticed observation (means this value increase or decrease). What I would like to underline is the fact that the interpretation of the results should start firstly, with the what the respective figure illustrate and then, about the presentation of the results (means obtained values) and after, the comparison/description (interpretation). Please provide to the results section the required corrections in order to allow the reader to follow the idea. The results are presented more as form of discussion.
Response 2: We have added additional detail to the results, including more quantitative values. This includes citing the value first and then the observation. We now more precisely describe the data and remove any discussion in the results.
Point 3: Based on the valuable results, is mandatory to design both: the scheme representing shortly step by step the present work as well as the mechanism mediation on NRK 52E and human HK-2 and RPTEC (in vitro). This aspect will support the main hypothesis proposed in present study. Then, when the author when will prepare the additional studies will design the mechanism in vivo.
Response 3: We are not sure what the reviewer is referring to here. Is this additional detail as requested in 2 above?
Point 4: Material and methods section: please supplement the Flow cytometry study with the used parameters (means cells/sec., etc. ). Also the information regarding the threshold. Did the author performed the threshold step? How the cells of your interest have been measured using fluorophores for investigation of early, late apoptosis and necrosis (means the way from the receiving ssc/fsc to characteristic scatterplot). Have been excluded from the results interpretation the scattered plots that are not of your interest? Please provide and explanation to the materials section
Response 4: We added this information, as well as our method of gating out debris, to the methods section. We did not exclude any cells from the interpretation as such data can bias results when studying cells death. We did not analyze any events under 200 in the SSC and FSC, as these are not cells but debris. More detail is presented in the methods. The methods used to assess annexin and PI staining, as well as cell cycle, are described in several of other studies, and our section reflects the same level of detail as these other studies.
Point 5: Table 1: the quality of the compound structures from the table should be improved. Figures 1, 3-5, 8, 9: the resolution of the figures must be improved. Especially, it is difficult to recognize the axis x/y name from the histograms received from the flow cytometry analysis. Also Figure 5, the statistical data are really unreadable. Please not remove but improve. All represented info is valuable.
Response 5: Thank for this. We have increased the size and quality of all figures.
Point 6: Legends also should include clearly the information what represent. For example Fig 4: “Effect of BFRs on the cell cycle in (a) NRK cells and (b) RPTECs after 24 hours of exposure to the IC50. Each bar represents the mean ± SD percentage of cells undergoing each phase of the cell cycle across n=3 passages of cells” – the respective legend did not present clearly what illustrate. Should be include representation of the bar graph as well as the histograms, representing what?......in this context please revise all figures legends.
Response 6: This information is now included.
Round 2
Reviewer 2 Report
I would like to thanks the authors for the engagement and clear answer following the Reviewer’s comments and suggestion. However, some aspects recommended by the Reviewer, have not been considered. One of them, regarding the scheme that design the study and action mechanism that may take place during the investigation of the cytotoxicity, has not be understood, therefore, I will make an additional effort and try to detailed my meaning. So, the present study have shown valuable results and the designing of the mechanism action will make the present research more interesting and will give relevant aspect.
The author studied molecular mechanisms of BFR toxicity in rat and human kidney cells after acute exposure to three BFRs for the first time based on RNA sequencing. In this fact, I would like to recommend to design the mechanism action, means, scheme, representing graphically all the information presented. How these investigated three BFRs influence the mechanism action on rat and human kidney cells. This aspect, will allowed the researchers to follow easer the idea and on the another hand will supplement/compliment the present knowledge in this context.
Also I will make additional effort to ask the author to improve the quality/resolution of the Fig 3 and 4, especially scatters plots and histogram. It is very difficult to read the x and y axes. Please present them in clear way.
Moreover, it is difficult to follow the Fig 5 B, C (Principal components analysis), the name of the samples are presented unclear and also no information in the legends regarding the additional explanation of the used color has not been presented.
Please follow step by step additional comments from the Reviewer.
Author Response
Reviewer 2 comments:
The author studied molecular mechanisms of BFR toxicity in rat and human kidney cells after acute exposure to three BFRs for the first time based on RNA sequencing. In this fact, I would like to recommend to design the mechanism action, means, scheme, representing graphically all the information presented. How these investigated three BFRs influence the mechanism action on rat and human kidney cells. This aspect, will allowed the researchers to follow easer the idea and on the another hand will supplement/compliment the present knowledge in this context.
Also I will make additional effort to ask the author to improve the quality/resolution of the Fig 3 and 4, especially scatters plots and histogram. It is very difficult to read the x and y axes. Please present them in clear way.
Moreover, it is difficult to follow the Fig 5 B, C (Principal components analysis), the name of the samples are presented unclear and also no information in the legends regarding the additional explanation of the used color has not been presented.
Response to Reviewer 2:
We appreciate Reviewer 2’s valuable comments. We have improved the resolution of figures 3 and 4. Regarding the PCA plots, there is a color legend at the top of each plot that explains what treatment group each color represents. (Specifically, it lists the cell line and BFR for each treatment group). We have also added a note to the figure caption that explains this to the reader. Regarding further detail about the mechanism of action of these drugs: unfortunately, we feel that such a conclusion is beyond the scope of our data. By performing gene set enrichment analysis, we highlighted several diverse potential mechanisms of toxicity. A more in-depth description of the mechanisms of BFR nephrotoxicity can only be accomplished by future studies that follow up on our data by measuring specific markers of these GSEA pathways to figure out how they function in vivo. Our discussion section addresses the reviewer’s question while still considering the variety and scope of our data. Specifically, we selected a couple of the most significantly enriched GSEA gene sets and explained 1) whether these have previously been shown to be involved in BFR toxicity in other target organs, and 2) whether these are involved in the mechanism of nephrotoxicity of other drugs. We sincerely hope that the reviewer understands why we cannot choose one mechanism and explain it in more depth based on the data shown.